

**Assessment of seawater intrusion using multivariate statistical, hydrochemical and**
**geophysical techniques in coastal aquifer, Cha-am district, Thailand**
Jiraporn Sae-Ju[1], Srilert Chotpantarat[1, 2, 3, 4], and Thanop Thitimakorn[1,3]
[1] Department of Geology, Faculty of Science, Chulalongkorn University, Bangkok 10330, Thailand

[2] Center of Excellence on Hazardous Substance Management (HSM), Chulalongkorn University,

Bangkok, Thailand.

[3] Research Program on Controls of Hazardous Contaminants in Raw Water Resources for Water Scarcity

Resilience, Center of Excellence on Hazardous Substance Management (HSM), Chulalongkorn

University, Bangkok, Thailand.

[4] Research Unit of Green Mining (GMM), Chulalongkorn University, Bangkok, Thailand.

[*] **Corresponding author. Tel.**: (662)2185442; **Fax.**: (662)2185464; **Email**: csrilert@gmail.com.



**ABSTRACT**
Seawater intrusion in coastal areas is one of the important environmental problems, causing negative impact
on groundwater resources in the future. To assess and mitigate the seawater intrusion, the affected aquifers
need to be characterized. By integrating geophysical investigation and multivariate statistical analysis of
the hydrochemical data, seawater intrusion into coastal aquifers in this area could be evaluated. The study
conducted 80 locations of the vertical electrical sounding (VES) survey; then selected 47 VES to create
four pseudo cross-section lines in a west-east direction, running perpendicular to the coast in Cha-Am
district, Thailand, which is negatively affected by this problem. The geophysical results were described
together with the hydrochemical analysis of 57 groundwater samples. The results revealed that seawater
intrusion occurred in the Qcl aquifer with an average depth of 50–60 meters and presented more obviously
near the coastal line. The resistivity value of $<5$ $\Omega$m represented highly contaminated areas impacted by
seawater intrusion while the range between 5–10 $\Omega$m represented moderately contaminated areas.
According to the hydrochemical characteristics, groundwater can be divided into three groups according to
the level of impact of seawater intrusion: Ca-Na-HCO$_3$ and Ca-HCO$_3$-Cl (slightly impacted), Ca-Na-HCO$_3$-
Cl (moderately impacted), and Na-Cl (highly impacted). The area had a low resistivity value, corresponding
to the high value of electrical conductivity (EC), and the hydrochemical facies was generally Na-Cl. The
hydrochemical facies evolution diagram (HFED) revealed that most of the samples fell close to the mixing
line, demonstrating mixing between seawater and fresh water and that some samples fell in the intrusion
phase. According to multivariate statistical analysis, the finding was in agreement with the HFED. There
are three main processes: seawater intrusion, natural groundwater recharge, and finally hydro-geochemical
interaction. Finally, the findings in this study demonstrated that the levels of seawater intrusion could be
classified into three zones depending on the degree of seawater intrusion. Furthermore, the northern part of
the study area faced seawater intrusion with a relatively higher impact than other areas, and seawater
laterally intruded about eight kilometers inland.
**Keywords:** coastal aquifer; hydrochemical facies evolution diagram; hydrogeological characteristics;
seawater intrusion; Thailand; vertical electrical sounding method



# 1. INTRODUCTION

Many coastal areas in the world contain densely populations as it is an area that has food integrity and important economic activities such as urban development, trade, and touristic activities. These are factors that have attracted people to settle in these areas, as a result, the water demand for consumption, agriculture, and industry has increased. Groundwater resources are an alternative water source. Compared with surface water, the groundwater is of high quality, is barely affected by seasonal effects (i.e., constant temperature), with large available quantities. For the reasons above-mentioned, groundwater has long been pumped with a large quantity of water. As a result, the common phenomenon, so-called seawater intrusion, has occurred in many coastal areas worldwide (Mas-Pla et al., 2014; Shi and Jiao 2014; Morgan and Werner, 2015).

In the coastal aquifer, seawater lies under fresh water since fresh water is less dense than seawater; consequently, the zone of contact between fresh water and seawater is brackish water (Bear, 1999). Fresh water is commonly over the top of the heavier seawater and serves to push the seawater interface seaward. In contrast, when pumping fresh groundwater from coastal aquifers with a large quantity, the pressure of fresh water is reduced, which in turn causes the seawater to migrate further landward. The seawater intrusion problem is one of the most important environmental issues that negatively affects groundwater resources significantly since groundwater salinity can lead to a reduction in fresh water availability and the degradation of groundwater quality (Essink, 2001; Werner et al. 2013; Kang and Jackson 2016; Ros and Zuurbier 2017). Therefore, the study of seawater intrusion into coastal aquifers is needed to identify the affected zones where it should be able to prevent problems or remediate such areas efficiently.

The coastal study area is located at Amphoe Cha-am, Changwat Phetchabur, and was selected as it is considered a densely populated area and is one of the most famous tourist areas in Thailand. Therefore, the groundwater resource may become a primary water resource in the near future and is consequently drawn out over the yield in the aquifer. Under this current situation, the natural equilibrium of the seawater interface is directly changed, and the sea water laterally moves landward. Geophysical and hydrochemical techniques have been integrated to investigate areas disturbed by seawater intrusion (McInnis et al., 2013;





Agoubi et al., 2013; Kouzana et al., 2010; Cimino et al., 2008; Song et al., 2006; Kazakis et al., 2016; Fadili et al., 2017; Najib et al., 2017). Geophysical techniques such as the 1-D electrical resistivity survey or vertical electrical sounding (VES) have been used since the electrical resistivity between fresh water and seawater saturated zones show large differences (Van Dam & Meulenkamp, 1967; Sabet, 1975), therefore, it is capable of identifying the contrast in terms of resistivity values between seawater and freshwater in coastal aquifers. In addition, the VES technique can be enabled in large areas as it is less time-consuming and has an economical cost when compared with drilling exploration methods. However, geophysical survey data are not capable of identifying the clearly lateral penetration of seawater in various lithologic units in a hydrogeological formation and groundwater facies or chemical constituents as the low resistivity depends on various factors such as the formation materials and groundwater chemistry (Zohdy et al., 1974). As a result, to fulfill the limitations of geophysics for delineating seawater intrusion areas, the integration of a hydrogeological investigation (with the help of lithologic information from drill wells), chemical analysis of groundwater samples, and multivariate statistical analysis were carried out. Therefore, the objectives of this study were to integrate multiple techniques including multivariate statistical, hydrochemical, and geophysical approaches to delineate the impact of seawater intrusion in coastal aquifers in the study area and further explain the geo-hydrochemical process in the coastal aquifers.

## 2. STUDY AREA

The study area is located in Amphoe Cha-am, Changwat Phetchaburi, which is a part of the central part of Thailand. The area covers approximately 360 square kilometers and lies between the latitudes 12°37.6'–12°53.845' N and longitudes 99°50.827'–99°729' E (Figure 1). The area is bounded by the northern and western borders of Tha-Yang District, by the southern border of Prachuap-Khiri-Khan Province, and by the eastern border of the Gulf of Thailand (Figure 1a). The area can be classified by topography into two major landforms consisting of a plain interleaved mountain covers (~20% of the area) and low-plain or coastal plain covers (~80% of the area). The plain interleaved mountain landform is located to the west of the area, covered mainly by forest areas, while the eastern side is a low-plain or coastal plain



to the Gulf of Thailand, and consists of various land use types including agriculture, facilities, forests, and
water bodies, respectively (Land Developement Department, 2011) (Figure 1b).

*[Insert Figure 1]*

*2.1. Geological Setting*
The study area is located on the Shan-Thai subcontinent consisting of Carboniferous rocks to
Quaternary sediments as shown in Figure 2. The Permo-Carboniferous sedimentary rocks and Permian
limestone are the basement rocks in the area, and are found distributed to the east of the study area. In
addition, basement rocks intruded by Cretaceous granite that appear as a mountain range and isolate hills
are found in the west of the study area. The basement rocks are filled with Quaternary sediments.
Quaternary sediments consist of marine sediments that are coastal tide-dominate deposits, colluvial
sediments that accumulate around the hill foot (Department of Mineral Resources, 2014).

*[Insert Figure 2]*

*2.2. Hydrogeology*
The study area is underlain with unconsolidated and consolidated aquifers (see Figure 3). More
than 60% are underlain with the Quaternary unconsolidated aquifer that consists of a beach sand aquifer
(Qbs). These are common sediments in the east coastal plain and are distributed from north to south.
Therefore, the groundwater has accumulated in the pore space of the sand deposited in the old ridge. The
average depth of the aquifer is 5–8 meters with groundwater levels of approximately 1–2 meters deep. Well
yield is less than 2 m³/hour. The total dissolved solid content (TDS) ranges from 500–1000 mg/L. The flood
plain aquifer (Qfd) is dispersed over in the upper part and is lined between Qbs and granitic aquifers (Gr).
The groundwater has accumulated in the pore space between the gravel and sand grains. The average depth
of the aquifer is 25–45 meters with groundwater levels of approximately 3–8 meters deep. Well yield is



between 2–10 m³/hour, but in some areas, its range from 10–20 m³/hour. The TDS ranged from 600–2000
mg/L. The colluvial sediments aquifer (Qcl) deposits near the foothill of the granite mountain are in the
north and south of the area. Thus, the sediments are poorly sorted, are angular to sub-angular, and the
groundwater has been stored in the pore spaces of the gravel, sand, quartzite fragments, granite fragments,
and clay deposits. The average depth of the aquifer is 30–50 meters. Well yield is relatively high at
approximately >50 m³/hour. The TDS ranged widely from 300 to >1200 mg/L. The consolidated aquifers
are composed of 40% of the study area as described follows. The Silurian-Devonian metamorphic aquifer
(SDmm) was found in a small area in the lower west. The average depth of the SDmm aquifer is 30–35
meters with a groundwater level of approximately 3–5 meters deep. Well yield ranged from <2 m³/hour.
The quality of the groundwater is generally good with a TDS <800 mg/L. The Permian-Carboniferous
metasedimentary aquifer (PCms) can be classified in two groups. First, in the northern part, groundwater
has accumulated in the fracture of mudstone and shale inter bedded with sandstone. The average depth of
the aquifer ranged from 50–200 meters. Well yield ranged from <2 m³/h, but some areas can develop
groundwater of 5–10 m³/h in the fracture zone. Second, in the upper eastern part of the study area,
groundwater has accumulated in the fracture of quartz rich sandstone interbedded with thin shale. The
average depth of the aquifer is 50–100 meters. Well yield ranged from 2–10 m³/h, but some areas develop
groundwater of >20 m³/h. The Permian limestone aquifer (Pc) has been found dispersing a small
monadnock in the north of the area. The average depth of the aquifer is 10–50 meters with a groundwater
level of approximately 3–5 meters. Well yield ranged between 10–20 m³/h. The quality of groundwater is
good with a TDS of <500 mg/L. Finally, the Cretaceous granite aquifer (Gr) is dispersed along the large
mountain ranges in a north-south direction to the west of the area. The average depth of the aquifer ranged
from 50–150 meters with groundwater levels of 6–9 meters deep. Well yield is <2 m³/h, but some areas can
develop groundwater of more than 20 m³/h (in fracture and/or fault zones). The TDS ranged from 200–
1000 mg/L (Department of Groundwater Resources, 2015).

*[Insert Figure 3]*



### 3. METHODOLOGY

Seawater intrusion into coastal aquifers can be detected through several methods. In most of the case studies, the electrical method has been universally recognized (George et al., 2015; Cimino et al., 2008; Samouëlian et al., 2005) as the electrical conductivity (EC) of a saturated subsurface depends primarily on three major factors: porosity, the connectivity of pores, and the specific conductivity of water in the pore (Telford et al., 1990). The difference between the EC of the seawater saturated subsurface and freshwater saturated subsurface is significant; thus, the electrical resistivity survey was well suited for evaluating the relationship between freshwater and seawater in coastal areas (Sherif et al., 2006). The electrical method analyzed with hydrogeological measurements can map interfaces of fresh water and seawater in coastal aquifers more precisely (Zarroca et al., 2011). In this study, we used both the direct method by sampling groundwater samples for hydrochemical analysis, and the indirect method by Vertical Electrical Sounding (VES). Both the sampling of groundwater samples or VES investigation collected data in the dry season (April to May) since seawater intrusion may occur evidently.

*3.1. Vertical Electrical Sounding Method*

One-dimensional resistivity survey, known as VES, was applied in this research. The principle of VES is currently released into the ground through current electrodes A and B, then the voltage is measured by potential electrodes M and N. The resistivity meter converts the voltage to the resistance value, then is plotted on log-log paper in the field to check for anomalies. Eighty VES points in this research used the resistivity meter (Syscal R1 Plus model, Iris) by the Schlumberger configuration (Figure 4).

Figure 1a shows 80 VES points which can be divided into four profiles: A-A'; B-B'; C-C'; and D-D'). Four profiles were in a W-E direction oriented perpendicular to the coast line. The current electrode spacing (AB) and potential electrode spacing (MN) was based on a relationship of 2(AB/2) > 5 MN at all depths (Telford et al., 1990). The current electrodes (A and B) spacing were measured in meters and varied from 1 to 200 meters and the potential electrodes (P1 and P2) varied from 0.25 to 20 meters. The selected current electrodes spacing were: 1, 1.5, 2, 3, 5, 7, 10, 15, 20, 25, 30, 35, 40, 45, 50, 60, 70, 80, 90, 100, 110, 125, 135, 150, 160, 175, 185, and 200 meters. Apparent resistivity data derived from the field was



interpreted using IPI2WIN software version 3.1.2c developed by Moscow State University (Bobachev,
2003). The data from IPI2WIN were the inversion of the apparent resistivity measurement dependent on
the calculation, and the calibration field curves that are compared with the theoretical curves of the program.
The results of the VES survey method were the resistivity of the soil and rock layers vertically. The shape
and slope of the VES data on the graph represents changes of the layer that had different resistivity (Telford
et al., 1990). Therefore, the correct interpretation of VES required the use of information regarding the local
geology and drilling information of the groundwater wells.

*[Insert Figure 4]*

*3.2. Chemical Analysis of Groundwater*
A total of 57 groundwater samples were collected in the study area. Prior to sampling, the
groundwater must be pumped out for 5–10 minutes to represent the groundwater in the aquifer. Samples
were necessary to measure physical and chemical parameters such as pH, electrical conductivity (EC),
groundwater, and total dissolved solids (TDS) in the field by using portable meters. Samples for the analysis
of cations and anions were collected in two bottles of 500 mL Poly-Ethylene (1 bottle with nitric acid
($HNO_3$) added to maintain the acidity of the water and the other with no nitric acid to be used for anion
analysis. Samples were kept at 4 °C to reduce the process of microorganisms and reduce the speed of
physical and chemical processes. The groundwater analysis was divided into two parts: anion analysis and
cation analysis. Chemical analysis was carried out for a group of cations ($Ca^{2+}$, $Mg^{2+}$, $Na^+$, $K^+$, and $Fe^{2+}$)
analyzed by the Absorption Spectrometry Method (AAS), whereas the group of anions ($Cl^-$, $Br^-$, $NO_3^{2-}$, and
$SO_4^2$ ) were analyzed using the Ion Chromatographic Method (IC) and $CO_3^{2-}$, $HCO_3^-$ were analyzed by the
volumetric titration method. The results of the chemical analysis can be analyzed by the piper diagram and
hydrochemical facies evolution (HFE) diagrams to classify the types of water that seawater intrusion can
be indicated by an increase in TDS and the major ions of seawater.



Furthermore, correlation analysis and principal component analysis (PCA) were used to clearly
acquire the relationship among the hydrogeochemical parameters measured in the groundwater samples,
which enabled the identification of the hydro-geochemical processes occurring in the groundwater system.
In this study, the multivariate statistical technique were carried out using the SPSS software, version 22.
Ten hydrochemical parameters (K, Na, Fe, Ca, Mg, F, Cl, Br, $SO_4$, and $HCO_3$) as well as pH, and electrical
conductivity (EC) in the groundwater samples were carried out using correlation analysis and PCA. These
two techniques are a statistical technique to group and establish the relationship between a group of
groundwater samples based on hydrochemical characteristics (Abderamane et al., 2012).

## 4. RESULTS AND DISCUSSION

### 4.1. Vertical Electrical Sounding

Figure 1a shows 80 VES points. All 80 of the VES data showed the H type curve (Telford et al.,
1990) (Figure 5a). This implied that there were three layers in the subsurface that consisted of resistivity
with $\rho_1 > \rho_2 < \rho_3$ ($\rho_1$ = resistivity of upper layer, $\rho_2$ = resistivity intermediate layer, and $\rho_3$ = resistivity
bottom layer) and were consistent with the characteristics of geology and hydrogeology of study area. The
top layers represented the unsaturated sediments, the middle layer was the saturated sediments, and the
bottom layer was interpreted as bed rock (Figure 5b).
***[Insert Figure 5]***
Figure 5b shows the results of the interpretation of VES St25 where apparent resistivity is the black
line and the master curve is the red line. It was found that at the St 25 location, it could be divided into four
layers (blue line). The top layer had a resistivity of 9.38 Ωm with a thickness of 3.11 m and a depth of 3.11
m. The second layer had a resistivity of 2.02 Ωm, thickness of 11.3 m, and depth of 14.4 m. The third layer
had a resistivity of 1.49 Ωm, thickness of 60.1 m, and a 74.5 m depth. The bottom layer was the bedrock
layer with a resistivity of 757 Ωm. The RMS error was 2.46%. After the layers were interpreted for all
points, then pseudo cross-sections for the four lines were generated.



The pseudo cross-sections (A-A', B-B', C-C', and D-D') were generated from 1D VES data that
selected 47 VES points from this study and 9 VES points from the Department of Groundwater Resources
(DGR) database. All cross-section lines were created from west to east that was perpendicular to the coast
line (Figure 1a).
The pseudo cross-section A-A' (Figure 6a) had 14 VES that consisted of St1, St2, St3, 42–28, St4,
F5, 42–29, St5, St6, F6, St7, St8, St9, and St10. This line was located at the lower part of the study area
and the total length was approximately 6750 m. It was found that the surface layer (0–10 m) had resistivity
ranging from 10–1000 Ωm. It was interpreted as an unsaturated zone of the Qfd aquifer. At the western side
of the profile, very high resistivity (400–1000 Ωm) was observed. This is due to the granite batholith that
intrudes in the base rock, as a granite outcrop was found near the St8. In the eastern side, at a depth of 130–
200 m, the layer that had a resistivity of 50–100 Ωm was the bedrock layer. The top of this side showed
resistivity values between 3–40 Ωm, and represents sediment saturated with water, especially near the
coastal line where there was a very low resistivity zone (<5 Ωm). This may represent the influence of
seawater intrusion that intrudes inland about one kilometer from the coast line.
The pseudo cross-section B-B' (Figure 6b) consisted of 16 VES points (St11, St12, St13, St14,
St15, St16, St17, St18, St19, St71, St20, St21, St22, St23, 42–126, and 42–125) and started from east to
west. This line was close to the A-A' profile and total distance was approximately 8450 m. It was found
that the surface layer (0–10 m) had resistivity anomalies between 50–400 Ωm. This represents the
unsaturated zone of the Qfd aquifer. The next layer showed resistivity values ranging from 2–20 Ωm. This
was interpreted as a sediment layer saturated with water. Near the coast line, there was a low resistivity
zone (<5 Ωm) that extended approximately two kilometers inland, which may represent the influence of
seawater intrusion. At the west side of the profile at a depth 20–200 m, there was a zone of resistivity
ranging from 60–250 Ωm that represented the bedrock layer.
The pseudo cross-section C-C' (Figure 6c) consisted of 12 VES as follows: St24, St25, St26, St27,
St28, St29, St30, St31, 41–205, St32, 41–191, and St33 from the east to west direction. This profile was
between the B-B' and D-D' profiles. The total distance of this line was approximately 7975 m. The surface



(0–10 m) layer had a resistivity range from 6–700 Ωm, which represented the unsaturated zone of Qfd
aquifer. The next layer showed resistivity values between 2–20 Ωm and represents the sediment layer
saturated with water, and there was a low resistivity zone (<5 Ωm) near the coast line where it extended
inland approximately 3.2 kilometers, representing the influence of seawater intrusion. Furthermore, the
western side of the profile at a depth 40–200 meters showed resistivity values between 60–200 Ωm,
representing the bedrock layer.
The pseudo cross-section D-D' (Figure 6d) consisted of 14 VES as follows: St34, St35, St36, St37,
St39, St41, St42, St43, St44, 41–174, 41–173, St45, St46, and St47 from an east to west direction. This
profile was on the northern part of the study area and the total length was approximately 14,300 m. It was
found that the surface (0–10 m) layer had resistivity in a range from 1.5–400 Ωm, representing the
unsaturated zone of the Qfd aquifer. In this profile, there were low resistivity anomalies (<5 Ωm) near the
surface at St7, possibly caused by the waste water from community areas. The next layer showed resistivity
values in a range from 1.5–20 Ωm, representing the sediment layer saturated with water. Near the coast
line, the low resistivity zone (<5 Ωm) extended far from the coast inland approximately four kilometers,
representing the influence of seawater intrusion. In the west side of the profile, the bedrock was found with
resistivity values of 40–100 Ωm at a depth between 90–200 meters.
Since soil or rock layers with low resistivity may be caused by many reasons including the clay
content or salt water in hydrogeologic formations, this research compared the lithologic data with the VES
data survey and found that the low resistivity at the area near the coast line may have been caused by the
seawater. Figure 7 shows the resistivity values compared with the litho-log of lines D-D'. The left side
shows the lithological data of well no. Q168 versus the VES of St47 that represents a position located far
from the coast, and the right side shows the lithological data of well no. PW7962 versus the VES of St34
that represents a position located near the coast. It was found that the low resistivity (<5 Ωm) at VES St34
corresponded to a gravelly clay layer, but at VES St47, the layer with components of clay or clayey gravel
showed a resistivity of only 10–20 Ωm. From the comparison, it can be concluded that although the layer
had a clay component, the resistivity value was not very low (<5 Ωm). As a result, the area that showed



very low resistivity may have been influenced by seawater intrusion. This is consistent with previous studies
(Ravindran, 2013; Kaya et al., 2015; Gopinath and Srinivasamoorthy, 2015) who concluded that the areas
with low resistivity (<5 $\Omega$m.) were influenced by seawater intrusion.

*[Insert Figure 6]*
*[Insert Figure 7]*

*4.2. Geoelectrical Section*

When the pseudo cross-sections of the four profiles were analyzed with borehole log and electric

log data, the comparison of these two data was established as shown in Figure 8. The four geo-electrical
cross-sections clearly showed the boundaries of each aquifer. The top layer was the Qfd aquifer consisting
of well sorted sand with high sphericity of approximately 0–20 m thick. The next layer was the Qcl aquifer,
which consisted of clayey gravel (poorly sorted and angular to sub-angular) interbedded with sand in some
areas. The average thickness was approximately 50–60 m, but it gradually increased by 100 meters in the
area near the coast line. The next layer underneath the Qcl aquifer was the PCms aquifer, consisting of
greenish gray sandstone and shale with an average depth between 50–200 meters. In the A-A' and B-B'
profiles, which were located in the lower part of the study area, the PCms aquifer was not found, whereas
granite (Gr) aquifers could be found in all profiles. In the D-D' profile, the Pr aquifer was located in the
central part of profile. It lay on top of the PCms aquifer at an approximate depth of 50–60 m. Since this
area had a limited number of borehole log data, the interpretation to create a cross-section in some areas
was required as shown in the dashed line in the cross-section.

*[Insert Figure 8]*

*4.3. Apparent Resistivity Map*



Figure 9 shows the apparent resistivity map established from the apparent resistivity of VES for
comparison of the resistivity of various depths. In this study, the apparent resistivity values for AB/2 = 5,
10, 30, 50, 70, 100, 150, and 200 meters were selected to create an apparent resistivity map by using the
ArcGIS9.3 program. The dark blue color on the map represents low resistivity and the red color represents
high resistivity, respectively. Figure 9 shows that all depths showed the location of low apparent resistivity
(<5 Ωm) in the same place. This was located on the east side of the study area which covered Tumbon Tha-
Yang, Nong-Sa-La, Bang-Kao, and some parts of Tambon Cha-Am. The maps of AB/2 = 5 and 10 meters
(Figures 9a and 9b) showed low apparent resistivity distributed in a sporadic pattern. This was found in a
wide area in the maps of AB/2 = 30, 50, and 70 m and was decreased in maps of AB/2 = 100, 150, and 200
m (Figure 9). The area with the highest apparent resistivity (>1000 Ωm) was located on the western side of
the study area covering most areas of Tumbon Khao Yai and Sam Pra Ya, corresponding to the bedrock in
the area. The map of AB/2 = 70 meters (Figure 9c) had the widest dark blue (<5 Ωm) area, which
corresponded to the depth of 50 m from the ground surface when compared to the cross-section;
consequently, it was found that this depth was located in the Qcl aquifer. As a result, it can be concluded
that the Qcl aquifer has been highly influenced by seawater intrusion.

*[Insert Figure 9]*

*4.4. Groundwater Chemistry*
The hydrochemical analysis of 58 groundwater samples is shown in Table 1. The total dissolved
solids (TDS) ranged from 195–3580 mg/L. The electrical conductivity (EC) of the groundwater samples
ranged from 292–5360 mS/cm. Figure 8 shows the hydrochemical facies classification of the groundwater
samples with the chemical result plotted on a piper diagram (Galloway & Kaiser, 1980) by the Groundwater
Chart Program from United States Geological Survey (USGS). The reliability of the results of the
hydrochemical analysis determined the charge balance of cation and anion (%Δ, which in this study was
acceptable at %Δ <10 (ALS Environmental). From Table 1, it was found that 42 samples were acceptable



(%Δ <10) and could be further used for discussion and interpretation while the unreliable results of 16
samples may have resulted from the process of groundwater collection and preservation, as well as the
dilution of the concentration of samples.

Figure 8 presents the piper diagram that shows the hydrochemical facies classification of 42

samples. The piper diagram can be divided into five facies as follows: the Ca-Na-HCO$_3$, Ca-HCO$_3$-Cl, Ca-
Na- HCO$_3$-Cl, Ca-Na-Cl, and Na-Cl facies represent fresh water and were found in W5, which was opened
as a well screen to the Gr aquifer. This groundwater facie is younger and is weakly acidic, which is generally
found in high terrain or recharge areas (Appelo and Postma, 2005). The Ca-HCO$_3$-Cl facie was found in
W19 and W20 that were opened as a well screen to the Gr aquifer. This facie was in water-bearing
permeable rocks given that when groundwater moves through the rock formation, the ion exchange process
takes place (Appelo & Postma, 2005). The quality of the water in this facie is fresh water. The Ca-Na-
HCO$_3$-Cl facie was found in 21 wells, which opened the well screen in four aquifers as follows: Gr aquifer
(eight wells), PCms aquifer (nine wells), Pr aquifer (one well, namely, W52), and Qcl (three wells). This
facie presents a complex chemical pattern since the water was influenced by many factors. This may have
resulted from the mixing of fresh water with seawater (Appelo and Postma, 2005). The Ca-Na-Cl facie was
found in W53 that was opened as a well screen to the Pr aquifer, while three wells (W24, W25, and W34)
were opened as a well screen to the Qcl aquifer, and two wells (W22 and W23) were opened as a well
screen to both the Qcl and PCms aquifers. The groundwater facies may have been changed due to the
influence of seawater intrusion by the ion exchange process (Appelo and Postma, 2005). The Na-Cl facie
was found in 10 wells that were opened as a well screen to the Qcl aquifer. The groundwater facie indicated
that these wells were influenced by seawater intrusion, and the quality of the water was saline water.

*[Insert Table 1]*


According to the piper diagram by Galloway and Kaiser (1980), the groundwater facie can be

classified into five facies, which correspond to the influence of seawater intrusion, and not only depend on



the distance from the coast, but also on the depth of the well or the screen level of the well. Most of the
groundwater samples in the Qcl aquifer were relatively influenced by seawater intrusion. In general, an
average depth of the Qcl aquifer ranged from 20–50 m and the thickness was typically higher than 100
meters approaching the coast line. The Na-Cl facies (approximately ~48%) and the Ca-Na-Cl facies
(approximately ~19%) were mainly found in the Qcl aquifer. The remaining approximate 20% of the
groundwater wells represented Ca-Na-HCO$_3$-Cl. According to the geochemical facies, the average
composition of cations (in meq/L) in the Qcl aquifer was in the following order: Na >> Ca > Mg~K.
Furthermore, the composition of anions (in meq/L) in the aquifer was in the descending order: Cl >> HCO$_3$
> CO$_3$ > SO$_4$. Therefore, their hydrochemical composition could be addressed as seawater intrusion in this
aquifer (Ahmed et al., 2017). The groundwater facies in the PCms and Gr aquifers, which opened a well
screen in the high weathering zone (underneath Qcl aquifer) at a depth of 40–60 meters was composed of
the Ca-Na-HCO$_3$-Cl facies, suggesting that the groundwater was moderately affected by seawater intrusion.
Moreover, the groundwater facies in the PCms and Gr aquifers (where a well screen opened at a depth of
60–80 meters) were the Ca-HCO$_3$-Cl and Ca-Na-HCO3 facies, suggesting that the groundwater was slightly
affected by seawater intrusion due to the mixing of recharge rainwater.
The relationship between the depth variation with the Cl and Na concentrations in each aquifer is
shown in the Supplementary Information (Figure SI.1). The chloride concentration was mostly found in
relatively high concentrations in the Qcl aquifer, especially in shallow to moderate depths (approx. 15 to
70 m depth). However, this relationship of salinity with depth was not well correlated in the Pcms and Gr
aquifers, implying that the groundwater in these aquifers were not highly impacted by seawater intrusion.
The red line on the piper diagram (Figure 10) is the line that shows the hydrochemical evolution of
the groundwater facies by the cation exchange reaction (Appelo & Postma, 2005). This line resulted from
two points plotted between the composition of seawater (blue point) and fresh water (red point) on the piper
diagram. By using this line, the hydrochemical results of the groundwater samples fall in the zone between
the blue and red points, representing mixed water occurring between the fresh water and seawater. The
groundwater facies close to the blue point were Na-Cl facies and Ca-Na-Cl facies, while the groundwater





facies close to the red point were Ca-HCO$_3$-Cl and Ca-Na-HCO$_3$. The groundwater facies found between
the blue and red points were the Ca-Na-HCO$_3$-Cl facies. As mentioned earlier, these were similar to the
study of Zghibi et al. (2014), who studied contamination in the Korba unconfined aquifer, which was
influenced by seawater intrusion. They showed a chemical analysis of water on the piper diagram and then
interpreted the results by creating the Theoretical Mixing Line (TML) of seawater and fresh water. They
found that groundwater showed paths of hydrochemical evolution along the TML line. The groundwater
facies can be changed from a Ca-SO$_4$ type to a Ca-Cl type to an Na-Cl type, and vice versa, from a Ca-SO4
type directly to an Na-Cl type, indicating that the chemical composition of groundwater is changed by a
cation exchange reaction.

*[Insert Figure 10]*

Figure 11 shows the relationship between the Na and Cl ions. The dominant ions in seawater are

Na and Cl, while the dominant ions in fresh water are Ca and HCO$_3$ (Appelo & Postma, 2005). Therefore,
the study of seawater intrusion has to focus on the dominant ions of seawater, which are Na and Cl ions.
The plotted graph comparing the Na and Cl ions found that it exhibited a strong correlation where the
groundwater samples fell on a 1:1 line with an $R^2$ of 0.941. This relationship suggests that both ions have
the same origin from seawater. The concentrations of Na and Cl depend on the degree of seawater intrusion
into the aquifer. From Figure 11, the groundwater can be divided into three groups. The first group, located
in the red circle, mainly consisted of samples in the Qcl aquifer that are severely influenced by seawater
intrusion. The groundwater types in this group showed Na-Cl and Ca-Na-Cl. The second group appeared
in the green circle, and consisted of groundwater samples from the PCms and Gr aquifers, which are
moderately influenced by seawater intrusion. The groundwater types in the second group showed Ca-Na-
HCO$_3$-Cl facies. The last group appeared in the blue circle and consisted of groundwater samples from the
deeper zone (when compared to the second group) of the PCms and Gr aquifers, which are slightly
influenced by seawater intrusion. The groundwater types in the last group showed Ca-HCO$_3$-Cl facies and


Ca-Na-HCO3 facies. These were consistent with the previous studies of Agoubi et al. (2013) and Zghibi et
al. (2014) that revealed the characteristics of groundwater chemistry influenced by seawater intrusion. They
found that groundwater influenced by seawater intrusion had dominant ions of Na and Cl. Therefore, when
plotting Na and Cl, both ions are well defined in the correlation where the groundwater sample falls on the
1:1 line, indicating that both ions come from the same source (seawater). Similarly, according to the ionic
ratio estimated from Cl/(HCO3 + CO3) and recommendation of Raghunath (1990), who suggested the ratio
of 2.8 as the threshold for indicating the saltwater intrusion, we found that more than 85% of  groundwater
samples in the  Qcl aquifer were moderately to highly contaminated due to seawater intrusion (Ebrahimi et
al., 2016).

*[Insert Figure 11]*

Furthermore, the study of Al-Agha and El-Nakhal (2004) found that by plotting the results of

groundwater samples on the piper diagram, phases of freshening and intrusion could be interpreted, but it
is difficult to recognize the sequence of facies in detail. Therefore, this research used the hydrochemical
facies evolution diagram (HFED) developed by Giménez-Forcada (2010) to describe the dynamic of
seawater intrusion. The percentage of major ions in the hydrochemical process associated with the dynamic
of seawater intrusion interface, consisting of $Ca^{2+}$, $Na^+$, $HCO_3^-$, $SO_4^{2-}$, $Cl^-$, was considered in the HFED.
Figure 12 shows that the red block (no. 4) is the Na-Cl facies (seawater), and the blue block (no. 13) is the
Ca-$HCO_3$ facies (fresh water). After plotting the percentage of ions on the diagram, it will generate a mixing
line between the fresh water and seawater to divide the phases of seawater intrusion. When the groundwater
samples appear above the mixing line, it represents the freshening phase, and if it falls below the mixing
line, this implies that it is during the intrusion phase. The results found that most of the samples fell close
to the mixing line (facies path 4–7–10–13), demonstrating mixing between seawater and fresh water. Some
samples fell in the intrusion phase (below the mixing line). In this initial phase, water gradually increases
the salinity along the facies path 13-14-15-16, which causes a reverse exchange, showing a Ca-Cl facies



(no. 16), which was not found in a groundwater sample in this facies. Finally, in this phase, water evolves
toward facies that are closer to seawater (Na-Cl facies) along facies path 16-12-8-4, and most samples in
this study were found in these groundwater facies. However, in this research, the groundwater sample fell
in Na-mixCl (no. 3), Na-mixHCO3 (no. 2), Na-HCO3 (no. 1), and MixNa-HCO3 (no. 5), which are
characteristic of the freshening process, and was not found in all groundwater samples.

Most samples that fell in the Na-Cl (no. 4) facies were groundwater samples collected from the Qcl

aquifer, and corresponded to the Na-Cl facies, which appeared in the piper diagram. This indicated that the
groundwater samples in the Qcl were severely influenced by seawater intrusion. Moreover, the groundwater
samples from the weathering PCms and Gr aquifers fell in the MixNa-Cl facie (no. 8) and MixCa-Cl (no.
12), which corresponded to the Ca-Na-HCO$_3$-Cl facies in the piper diagram, indicated a moderate influence
of seawater intrusion. In addition, the groundwater samples from the PCms and Gr aquifers at deeper levels
(depth of 60–80 meters) showed the Ca- HCO3-Cl facies in the piper diagram and HFED fell in the Mix
Na-Cl facie (no. 8) and MixCa-Cl (no. 12). Even though it fell in the facies of fresh water in the HFED, it
was close to the mixing line; therefore, it was slightly influenced by seawater intrusion. These were
consistent with the study of Ghiglieri et al. (2012), who used HFED for depicting the salinization processes
in the coastal aquifers in Italy. They found that the sample plotted on HFED followed the succession of
facies along the mixing line, indicating that seawater and fresh water were slightly mixed or that the ionic
exchange process occurred. Moreover, they used the HFED results in comparison with the EC contour lines,
showing that seawater had intruded quite far inland. Najib et al. (2017) highlighted the succession of
different water facies developed between the intrusion and freshening phases by analysis on the HFED. The
formation of Na-HCO3 facies, which characterizes the last facies of the freshening phase, followed the
succession of Na-Cl, MixNa-MixCl, MixCa-MixCl, MixCa-MixHCO3, and Na-HCO3. Moreover, the
obtained HFED results allowed us to extend the intrusion process in the Holocene groundwater and accept
the fresh water recharge such as meteoric water and lateral recharge from rivers (Liu, 2017).

*[Insert Figure 12]*



*4.5. Multivariate Statistical Analysis*

4.5.1. Correlation matrix

In this study, the Pearson correlation technique was applied to assess the relationship between various hydrochemical variables of the groundwater samples, which were measured in the field and analyzed in the laboratory (see Table 2). The highlighted bold demonstrates the significant relationship at $p < 0.05$. The high correlation of these parameters indicates that these cations/anions contributed to mineralization in the aquifer systems. Chloride (Cl) is highly correlated with Na Na-Cl, $r = 0.960$, ($SO_4^{2-}$) Cl-SO4, $r = 0.835$, and (Br) Cl-Br, $r = 0.857$, implying that the groundwater system was influenced by seawater intrusion (Askri et al., 2016). In general, the dissolution of halite may be investigated as the linear relationship between Na and Cl ions (Hem, 1985). The total dissolved solid (TDS) had a significantly positive correlation with EC TDS-EC, $r = 0.963$, (Na) TDS-Na, $r = 0.934$, (Cl) TDS-Cl, $r = 0.923$, (Br) TDS-Br, $r = 0.826$ ($SO_4$) TDS-SO4, $r = 0.775$ (and moderately correlated with Mg) TDS-Mg, $r = 0.669$, (Ca) TDS-Ca, $r = 0.435$. Similarly, the EC was positively correlated with (Na) EC-Na, $r = 0.910$, (Cl) EC-Cl, $r = 0.914$, (Br) EC-Br, $r = 0.779$, ($SO_4$) EC-SO4, $r = 0.769$, (as well as moderately correlated with ($Mg^{2+}$) EC-Mg, $r = 0.636$, and ($Ca^{2+}$) EC-Ca, $r = 0.425$. These good correlations with EC implied that the increase in salinity was caused from the groundwater mineralization (Moussa et al., 2011; Zghibi et al., 2012). Calcium ($Ca^{2+}$) had a moderately positive correlation with $Mg^{2+}$ Ca-Mg, $r = 0.583$, and was moderately correlated with bicarbonate ($HCO_3$) (Ca-$HCO_3$, $r = 0.422$), implying the dissolution of calcite and dolomite minerals from the geologic formation. Moreover, the good correlation between Mg and Na (Mg-Na, $r = 0705$) indicated that the ion exchange was likely occurs during seawater intrusion.

4.5.2. Principal components analysis (PCA)

The varimax method used to rotate the parameters in the principal component analysis uses the Kaiser criterion rotating with the varimax method. Therefore, all of parameters were classified into four components (see Table 3). Each component included the values (bold) that represent a good correlation. Table 4 shows the Eigen value of all parameters, and the four groups had Eigen values more than 1 and cumulative variance was more than 80.35%. A total dissolved solid (TDS), EC, $Mg^{2+}$, $Na^+$, $Cl^-$, $Br^-$, $SO_4^{2-}$



were included in the first component that had the highest factor loading (6.13) and accounted for 47.17%
of the total variance. This factor, with a high positive loading, ranged from 0.757 to 0.965, probably
indicating the consequence of seawater intrusion in the study area. Therefore, factor 1 can be defined as the
"seawater intrusion factor". Similarly, the study of Ahmed et al. (2017) found that factor 1, which was
defined as the seawater intrusion impact, accounted for 66% of the total variance and consisted of these
following elements: EC, Cl, Na, SO4, K, Mg, Br, Sr, B, Cr, Co, Arsenic, and Selenium. The second
component had a factor loading of 1.60, accounting for 12.28% of the total variance. The component
showed the positive relationship between $Ca^{2+}$ (0.62), $F^-$ (0.63), and $HCO_3$ (0.77). This factor can be
expressed as the natural process when recharge water infiltrates into the groundwater system and water-
rock interaction occurs, which eventually releases $Ca^{2+}$ and $HCO_3^-$ in groundwater (Jiang et al., 2009). The
values of $HCO_3$ widely ranged from 19.52 mg/L to 189.5 mg/L, depending upon various geological
formations in this area. Fluoride is naturally released from the dissolution of fluorapatite and flurite, which
occurs in sedimentary and igneous rocks. According to Rama Rao (1982) and Heinrich (1948), they
revealed that fluorite was detected in granite, gneiss, and pegmatites. Moreover, due to the similarity charge
and radius, fluoride can substitute the hydroxide ions via water-rock interaction. Similar to Ca and $HCO_3$,
fluoride (F) in groundwater can be inferred from the weathering of rocks in this area. The third component
consisting of $K^+$ and Fe showed positive loadings in the range of 0.7–-0.81 and the factor loading was
1.448 or 11.14% of the total variance. Through the weathering of igneous rocks, K can be mainly released
from potassium feldspar into the groundwater (Kim et al., 2004) and Fe represents the natural dissolution
of rocks and minerals via water-rock interaction. The last component consists of only pH showing the low
factor loading (1.27) or 9.78% of the total variance. These parameters might be the result of various dynamic
hydro-geochemical processes in the area such as seawater intrusion, recharge, water-rock interaction, etc.
In addition, all variables were plotted in rotated space in Figure 13 to clearly demonstrate the separation of
the four components.

*[Insert Table 2]*



*[Insert Table 3]*
*[Insert Table 4]*
*[Insert Figure 13]*

*4.6. Integrated Interpretation of VES and EC Results*

VES data has a limitation when further interpreting seawater intrusion as there are many factors

that show resistivity values of <5 Ωm in coastal aquifers. Therefore, the results of the hydrochemical
analysis and EC value needed to be interpreted with the VES data to overcome this limitation and to further
elucidate the seawater intrusion effect. The VES sections were overlaid on the EC map of the Qcl aquifer,
which was severely affected by seawater intrusion, then obtained the extent of intrusion as shown in Figure
16. The zone with resistivity values <5 Ωm were considered to be the seawater intrusion area. Furthermore,
the EC map showed the location of high EC (>1500 μs/cm), corresponding to the location of low resistivity
values (<5 Ωm).

From this relationship, it can be concluded that the Qcl aquifer was highly influenced by seawater

intrusion with resistivity values in the range of 0–10 Ωm, especially in the upper part of the area. The
boundary line is shown in Figure 14. Section A-A' was influenced by seawater intrusion about 3 km inland,
where the first kilometer from the coast line was highly influenced with resistivity values <5 Ωm, while the
last two kilometers were moderately influenced and represented brackish water with resistivity values
ranging 5–10 Ωm. Section B-B' was influenced by seawater intrusion about 5 km inland, where the first 2
km from the coastal line were highly influenced with resistivity values of <5 Ωm, while the last 3 km were
moderately influenced with resistivity values ranging from 5–10 Ωm. Section C-C' was influenced by
seawater intrusion about 4.7 km inland where the first 3.2 km from the coast line was highly influenced
with resistivity values of <5 Ωm, while the following 1.5 km were moderately influenced with resistivity
values of 5–10 Ωm. Section D-D' was influenced by seawater intrusion about 8 km inland where the first
4 km from the coast line was highly influenced with resistivity values of <5 Ωm, while the next 4 km was
moderately influenced with resistivity values of 5–10 Ωm. As shown from the geophysical and





535 hydrochemical results, the levels of seawater intrusion could be classified into three zones by using the

536 criteria shown in Table 5.


538 *[Insert Figure 14]*

539 *[Insert Table 5]*


541 **5. CONCLUSIONS**

542   In this research, 80 VES surveys were conducted using a Schlumberger configuration integrated

543 with the hydrochemical analysis of 58 groundwater samples to indicate seawater intrusion into coastal

544 aquifers. Four pseudo cross-sections were generated from the 80 VES data. These were in a good agreement

545 with those obtained from the hydrogeological data and lithologic data in the study area. The resistivity map

546 at different depths, generated from the VES data, successfully revealed the interaction of seawater and

547 freshwater along the coast line. The geophysical results found that seawater mainly intruded in the Qcl

548 aquifer. The resistivity values of <5 Ωm were found at a depth of approximately 50 m. However, the VES

549 is limited when evaluating the seawater intrusion in highly contaminated aquifer located close to the coast

550 line. Therefore, the evaluation of seawater intrusion in coastal areas with VES data needs the assistance of

551 hydrochemical and hydrogeological data to describe the seawater intrusion more accurately. According to

552 the hydrochemical analysis of 58 groundwater samples, five types of groundwater facies (Ca-Na-HCO3,

553 Ca-HCO3-Cl, Ca-Na-HCO3-Cl, Ca-Na-Cl and Na-Cl) were noticed, which were dependent upon aquifer

554 types and depths. Na-Cl facies were typically found in the Qcl aquifer, corresponding to the resistivity

555 values of <5 Ωm. As the geophysical and hydrochemical results showed, the levels of seawater intrusion

556 could be classified into three zones depending on the degree of seawater intrusion. As a suggestion of this

557 research, groundwater samples should be periodically collected from at least two periods to analyze the

558 dynamic and evolution of seawater intrusion. The VES investigation should be concerned with the distance

559 between the VES survey point and the VES point near the coast line as the depth of survey cannot penetrate

560 through the highly seawater contaminated groundwater. Moreover, the lithologic data for aquifers that have



a lot of clay components are required to interpret the results more accurately. In the future, when this area
needs to use the groundwater resources, people should use groundwater in the rock aquifers (PCms and Gr
aquifers) at depths of higher than 70 meters. The proper criteria for selecting the study area of seawater
intrusion should consider the following: groundwater demand in the area, ecological and hydrogeological
characteristics, and the amount of groundwater recharge needed to prevent problems that may occur from
a large amount of groundwater pumping in the future.

**ACKNOWLEDGMENTS**
Authors would like to thank the Geology Department, Faculty of Science, Chulalongkorn University, the
Graduate School of Chulalongkorn University, the International Research Integration: Chula Research
Scholar program, the Ratchadaphiseksomphot Endowment Fund (GCURS-59-06-79-01), the Office of
Higher Education Commission (OHEC), and the S&T Postgraduate Education and Research Development
Office ( PERDO)  for providing financial support for this research program. We are also grateful to the
Department of Groundwater Resources (DGR) for partially providing the data. We also thank the Editor
and anonymous reviewers for their suggestions and critical comments which have greatly improved the
earlier manuscript.












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

investigations. US Government Printing Office.











**Table 1** Hydrochemical analysis of groundwater samples

| Sample No. | Well No. | TDS (mg/l) | EC (µs/cm) | K (mg/L) | Fe (mg/L) | Ca (mg/L) | Mg (mg/L) | Na (mg/L) | F (mg/L) | Cl (mg/L) | Br (mg/L) | No3 (mg/L) | SO₄ (mg/L) | CO₃ (mg/L) | HCO₃ (mg/L) | %A |
|---|---|---|---|---|---|---|---|---|---|---|---|---|---|---|---|---|
| W1 | MU287 | 205 | 1000 | 12.25 | N/D | 35.4 | 5.6 | 95.8 | 0.347 | 82.12 | 0.839 | 0.64 | 7.16 | 25.2 | 51.24 | 23.105 |
| W2 | 5608C006 | 1307 | 1951 | 4.5 | 0.008 | 100.6 | 18.9 | 159 | 2.033 | 215.075 | 1.671 | 3.857 | 24.2 | 67.2 | 136.64 | 9.162 |
| W3 | 5408D026 | 773 | 1155 | 5.7 | N/D | 98.95 | 17.7 | 109.75 | 2.901 | 89.2 | 1.678 | 1.501 | 12.16 | 64.8 | 131.76 | 22.247 |
| W4 | MU747 | 607 | 905 | 8.4 | N/D | 48.4 | 13.3 | 82 | 1.184 | 128.92 | 1.368 | 0.353 | 0.708 | 39.6 | 80.52 | 7.185 |
| W5 | PCR9 | 718 | 1071 | 13.45 | N/D | 49.65 | 17.4 | 64.2 | 1.254 | 79.53 | 1.84 | N/D | 0.223 | 63.6 | 129.32 | 4.13 |
| W6 | C629 | 905 | 1348 | 16.05 | N/D | 48.01 | 18.2 | 129.6 | 1.195 | 219.525 | 1.535 | 7.9 | 13.2 | 22.4 | 72.3 | 5.202 |
| W7 | MU134 | 739 | 1105 | 14.75 | N/D | 41.9 | 9 | 115.5 | 0.823 | 229.5 | 1.475 | 0.308 | 0.054 | 18 | 36.6 | 3.402 |
| W8 | Private | 761 | 1137 | 11 | N/D | 45.3 | 12 | 137.75 | 1.806 | 215.4 | 2.098 | N/D | 1.763 | 18 | 118.6 | 4.783 |
| W9 | Private | 781 | 1170 | 7.7 | N/D | 58.95 | 12.9 | 40.2 | 3.18 | 84.84 | 1.94 | 2.889 | 4.928 | N/D | 132.6 | 9.927 |
| W10 | MU753 | 602 | 1123 | 9.15 | 0.147 | 17.6 | 8.7 | 54.6 | 0.134 | 119.88 | 1.065 | 0.033 | 9.939 | N/D | 48.8 | -2.149 |
| W11 | Private | 825 | 1360 | 17.45 | N/D | 46.9 | 18.5 | 127.25 | 1.239 | 216.475 | 1.919 | 10.52 | 9.615 | N/D | 141.52 | 2.452 |
| W12 | q1642 | 781 | 1166 | 13.35 | N/D | 69.4 | 20.5 | 115 | 1.265 | 167.6 | 2.566 | 7.367 | 5.736 | N/D | 119.56 | 17.754 |
| W13 | DCD14827 | 805 | 1230 | 6.55 | N/D | 70.6 | 34 | 103.2 | 1.462 | 169.975 | 2.008 | 1.449 | 16.3 | N/D | 112.24 | 21.629 |
| W14 | 5508C019 | 757 | 1130 | 4.75 | N/D | 58.95 | 34 | 92.2 | 1.192 | 164.94 | 2.791 | N/D | 2.402 | 52.8 | 107.36 | 9.137 |
| W15 | DCD14809 | 779 | 1162 | 6.8 | 0.548 | 68.75 | 19.7 | 90.6 | 0.833 | 180.95 | 2.228 | 0.149 | 0.046 | N/D | 145.9 | 9.978 |
| W16 | Private | 1650 | 879 | 43 | N/D | 86.15 | 24.6 | 241 | 0.513 | 336.28 | 3.654 | 28.88 | 44 | N/D | 158.6 | 8.641 |
| W17 | MU135 | 397 | 594 | 20.8 | N/D | 45.35 | 11.7 | 32 | 0.134 | 56.14 | 1.583 | 0.183 | 12.83 | N/D | 158.6 | 7.142 |
| W18 | Private | 1720 | 3024 | 4.75 | N/D | 168.4 | 27.8 | 199.6 | 3.115 | 400.88 | 3.589 | 10.64 | 84.96 | 60 | 122 | 4.461 |
| W19 | Private | 801 | 1196 | 14.55 | N/D | 70.3 | 18.2 | 40.5 | 1.25 | 72.8 | 2.996 | 9.478 | 12.32 | N/D | 182.7 | 8.826 |
| W20 | ND | 591 | 883 | 4.65 | N/D | 81.2 | 12.1 | 32.4 | 0.969 | 69.4 | 2.758 | 1.245 | 12.27 | N/D | 189.5 | 9.745 |
| W21 | 5408D022 | 782 | 1168 | 16.3 | 0.009 | 45.65 | 18.4 | 133 | 1.156 | 252 | 2.984 | 3.334 | 10.24 | N/D | 122 | 2.255 |





**Table 1** Hydrochemical analysis of groundwater samples (continue)

| Sample No. | Well No. | TDS (mg/l) | EC (µs/cm) | K (mg/L) | Fe (mg/L) | Ca (mg/L) | Mg (mg/L) | Na (mg/L) | F (mg/L) | Cl (mg/L) | Br (mg/L) | No3 (mg/L) | SO4 (mg/L) | CO3 (mg/L) | HCO3 (mg/L) | %A |
|---|---|---|---|---|---|---|---|---|---|---|---|---|---|---|---|---|
| W22 | DCD14772 | 556 | 830 | 8.95 | 0.103 | 50 | 13.8 | 107.6 | 2.728 | 204.78 | 2.198 | 0.034 | 4.215 | N/D | 102.48 | 6.208 |
| W23 | 5608C045 | 375 | 562 | 10.05 | N/D | 61.15 | 9.6 | 44.4 | 0.269 | 135.2 | 1.684 | 0.074 | 26.3 | N/D | 70.4 | 4.433 |
| W24 | MU563 | 900 | 1347 | 6.55 | 0.056 | 71.05 | 32.6 | 122.5 | 1.588 | 279.5 | 3.156 | 3.561 | 65.3 | N/D | 68.32 | 4.982 |
| W25 | MU564 | 843 | 1130 | 6.05 | N/D | 93.9 | 47.2 | 198.8 | 1.234 | 477.8 | 4.536 | 1.061 | 71.75 | N/D | 68.32 | 3.63 |
| W26 | 5408D011 | 195 | 292 | 14.1 | N/D | 15.65 | 4 | 33 | 1.944 | 34.6 | 0.836 | 12.02 | 1.287 | N/D | 21.96 | 13.368 |
| W27 | Q167 | 494 | 739 | 19.55 | 0.048 | 57.3 | 8.7 | 68 | 1.955 | 129.8 | 1.578 | 2.093 | 0.921 | N/D | 84.8 | 14.824 |
| W28 | q1607 | 391 | 584 | 12.7 | 0.879 | 45.95 | 9.8 | 62 | 1.934 | 125.6 | 1.664 | 0.183 | 0.031 | N/D | 94.6 | 9.056 |
| W29 | Private | 267 | 400 | 23.65 | 0.925 | 18.9 | 3.4 | 24.8 | 2.219 | 50.4 | 0.754 | 0.973 | 0.185 | 28.8 | 58.56 | -8.031 |
| W30 | MU283 | 420 | 627 | 16.35 | 0.209 | 45.4 | 7.1 | 42.9 | 2.154 | 87.6 | 1.541 | 0.567 | 6.204 | N/D | 124.6 | 4.608 |
| W31 | PCR10 | 460 | 687 | 9.05 | N/D | 45.15 | 11.7 | 87.6 | 2.492 | 100.5 | 1.68 | 0.072 | 8.604 | N/D | 70.76 | 26.942 |
| W32 | DCD14768 | 503 | 375 | 18.4 | 2.73 | 44.6 | 12.3 | 40.1 | 1.551 | 64.3 | 2.021 | 0.086 | 0.061 | 20.1 | 122 | 9.68 |
| W33 | MU348 | 507 | 757 | 10.05 | N/D | 56.3 | 13.1 | 70.2 | 1.308 | 130.15 | 1.946 | 5.401 | 3.659 | N/D | 132.5 | 6.632 |
| W34 | MU363 | 548 | 1015 | 8.65 | N/D | 51.4 | 13.1 | 118.2 | 1.199 | 259.6 | 0.953 | 1 | 8.577 | N/D | 62.3 | 2.377 |
| W35 | MU571 | 762 | 1138 | 15.65 | 0.069 | 45.6 | 17.2 | 134.6 | 1.001 | 269.8 | 2.659 | 1.949 | 9.96 | N/D | 82.6 | 3.334 |
| W36 | MU332 | 649 | 1002 | 57.3 | N/D | 59.35 | 21.8 | 82 | 0.288 | 178.6 | 2.225 | N/D | 24.2 | N/D | 112.24 | 14.043 |
| W37 | MU359 | 780 | 1267 | 58.3 | 1.104 | 58.75 | 17.8 | 123.4 | 0.199 | 247.95 | 3.282 | 0.316 | 29.25 | N/D | 97.6 | 9.942 |
| W38 | MU557 | 565 | 848 | 43.1 | 1.721 | 125.55 | 36.6 | 215.5 | 0.307 | 380.24 | 3.959 | 0.129 | 131.56 | N/D | 122 | 12.171 |
| W39 | MU458 | 239 | 357 | 15.1 | 0.387 | 15.5 | 5.9 | 33 | 0.17 | 64.3 | 0.812 | 1.052 | 2.281 | N/D | 34.6 | 10.366 |
| W40 | 5608C007 | 426 | 631 | 12.2 | N/D | 41.8 | 11.4 | 80 | 1.059 | 100.6 | 1.608 | 0.058 | 1.647 | N/D | 31.72 | 33.522 |
| W41 | MU361 | 559 | 834 | 11.2 | N/D | 49 | 13.6 | 87.6 | 4.325 | 97.3 | 1.711 | 4.303 | 24.43 | 51.6 | 104.92 | 4.512 |
| W42 | MU335 | 454 | 677 | 15.8 | N/D | 44.85 | 11.1 | 75.8 | 1.342 | 86.7 | 1.778 | 1.63 | 3.014 | N/D | 75.64 | 27.902 |
| W43 | MU549 | 621 | 929 | 14.15 | N/D | 76.1 | 18.9 | 93.8 | 1.975 | 154.3 | 2.084 | 8.536 | 5.307 | 51.6 | 104.92 | 7.02 |
| W44 | DCD14766 | 753 | 1124 | 12 | 0.007 | 58.35 | 16.4 | 82.4 | 2.31 | 170.425 | 1.806 | 0 | 2.524 | 52.8 | 107.36 | -1.354 |






**Table 1** Hydrochemical analysis of groundwater samples (continue)

| Sample No. | Well No. | TDS (mg/l) | EC (µs/cm) | K (mg/L) | Fe (mg/L) | Ca (mg/L) | Mg (mg/L) | Na (mg/L) | F (mg/L) | Cl (mg/L) | Br (mg/L) | No3 (mg/L) | SO4 (mg/L) | CO3 (mg/L) | HCO3 (mg/L) | %A |
|---|---|---|---|---|---|---|---|---|---|---|---|---|---|---|---|---|
| W45 | AFD8794 | 531 | 793 | 12.25 | 1.665 | 36.2 | 3.6 | 60.9 | 1.984 | 62.26 | 0.64 | 6.487 | 10.9 | 40.8 | 82.96 | -0.991 |
| W46 | MU588 | 254 | 379 | 8.25 | 0.019 | 24.75 | 7.8 | 32.4 | 0.743 | 15.486 | 1.05 | 0.042 | 5.706 | 16 | 54.2 | 27.687 |
| W47 | DCD14773 | 649 | 970 | 12.95 | N/D | 61.3 | 16.8 | 89.6 | 1.705 | 134.6 | 1.933 | 14.78 | 8.539 | N/D | 136.64 | 8.798 |
| W48 | MU578 | 629 | 939 | 17.05 | N/D | 70.9 | 18.3 | 85.2 | 1.564 | 53.18 | 2.169 | 16.88 | 10.12 | N/D | 87.84 | 35.666 |
| W49 | DOH11441 | 340 | 514 | 29.35 | 0.062 | 24.6 | 12 | 42.1 | 0.665 | 60.72 | 1.104 | 0.467 | 8.423 | 36 | 72.4 | 5.378 |
| W50 | MU331 | 660 | 986 | 20.95 | 0.066 | 66.25 | 22 | 92.4 | 0.771 | 184.2 | 2.135 | 0.356 | 3.86 | 39.6 | 80.52 | 9.848 |
| W51 | MU586 | 762 | 1138 | 13.3 | N/D | 72.9 | 12.9 | 84.3 | 1.96 | 171.1 | 2.244 | 2.601 | 4.256 | 14 | 122 | 7.023 |
| W52 | MU697 | 397 | 598 | 3.25 | N/D | 65.2 | 10.2 | 64.2 | 0.099 | 100.4 | 1.085 | 1.06 | 11.8 | 40 | 89.4 | 7.876 |
| W53 | q1605 | 2090 | 3120 | 4.85 | N/D | 214.55 | 30.5 | 288.4 | N/D | 920.9 | 4.776 | 9.142 | 74.8 | 36 | 73.2 | -8.29 |
| W54 | PCR16 | 3580 | 5360 | 14 | N/D | 38.05 | 37.8 | 665 | 0.247 | 1489.2 | 6.372 | 3.524 | 167.4 | 50.4 | 102.48 | -17.723 |
| W55 | q86 | 1920 | 2860 | 9.35 | N/D | 73.5 | 31.4 | 284.6 | 0.408 | 662.2 | 4.611 | N/D | 24.2 | 40.1 | 117.12 | -8.603 |
| W56 | q88 | 2920 | 4360 | 15.15 | N/D | 82.2 | 41.8 | 541 | 0.294 | 935.6 | 6.101 | N/D | 160.6 | 64.8 | 131.76 | -3.928 |
| W57 | C545 | 207 | 310 | 10.5 | N/D | 60.35 | 12.5 | 56.4 | 1.206 | 92.16 | 2.565 | 1.012 | 12.34 | 62.4 | 126.888 | -2.354 |
| W58 | Private | 784 | 1169 | 7.15 | N/D | 14.95 | 3.8 | 15.2 | 0.193 | 43.46 | 0.638 | 0.122 | 0.119 | N/D | 19.52 | 10.028 |
| Seawater | | 37675.2 | 53821.7 | 396.931 | 0.488 | 841.1 | 1083.43 | 11106.3 | 9.7944 | 21192.1 | 37.7172 | N/D | 2793.78 | 70.3998 | 143.142 | -2.78 |

\* Detection limit of IC standard (anion) = 10 ppm, AAS standard for $Ca^{2+}$ = 5 ppm, $Mg^{2+}$ = 0.25 ppm, $Na^+$ = 0.5 ppm, $K^+$ = 1 ppm and Fe = 3 ppm

\*\* Acceptable % error balance is less than ±10 % (ALS Environmental)

\*\*\*N/D = Not detected



**Table 2** Pearson's correlation coefficient )$R^2$ (between the physiochemical and hydro chemical parameter from 58 groundwater
samples.

| Variables | pH | TDS | EC | $K^+$ | Fe | $Ca^{2+}$ | $Mg^{2+}$ | $Na^+$ | $F^-$ | $Cl^-$ | $Br^-$ | $SO_4^{2-}$ | alkalinity |
|---|---|---|---|---|---|---|---|---|---|---|---|---|---|
| pH | 1.000 | | | | | | | | | | | | |
| TDS | -0.145 | 1.000 | | | | | | | | | | | |
| EC | -0.177 | **0.963** | 1.000 | | | | | | | | | | |
| $K^+$ | 0.032 | -0.035 | -0.118 | 1.000 | | | | | | | | | |
| Fe | 0.056 | -0.156 | -0.189 | 0.324 | 1.000 | | | | | | | | |
| $Ca^{2+}$ | -0.249 | **0.435** | **0.425** | -0.067 | -0.050 | 1.000 | | | | | | | |
| $Mg^{2+}$ | -0.227 | **0.669** | **0.636** | 0.052 | -0.093 | **0.583** | 1.000 | | | | | | |
| $Na^+$ | -0.196 | **0.934** | **0.910** | 0.046 | -0.097 | **0.362** | **0.705** | 1.000 | | | | | |
| $F^-$ | 0.216 | -0.187 | -0.169 | -0.295 | -0.010 | 0.038 | -0.164 | -0.237 | 1.000 | | | | |
| $Cl^-$ | -0.162 | **0.923** | **0.914** | 0.003 | -0.094 | **0.420** | **0.696** | **0.960** | -0.290 | 1.000 | | | |
| Br- | -0.072 | **0.826** | **0.779** | 0.115 | -0.048 | **0.564** | **0.849** | **0.847** | -0.223 | **0.857** | 1.000 | | |
| $SO_4^{2-}$ | -0.165 | **0.775** | **0.769** | 0.141 | 0.058 | **0.472** | **0.728** | **0.863** | -0.233 | **0.835** | **0.817** | 1.000 | |
| alkalinity | -0.104 | **0.378** | **0.342** | -0.046 | -0.015 | **0.422** | **0.346** | 0.282 | 0.222 | 0.206 | **0.399** | 0.229 | 1.000 |

Values in the highlighted bold indicate a relationship between two parameters with a significance level at 0.05 ($P<0.05$)





**Table 3**  Rotated component matrix dividing variables into four groups

| | Component | | | |
|---|---|---|---|---|
| | 1 | 2 | 3 | 4 |
| pH | -0.063 | -0.064 | -0.82 | **0.916** |
| TDS | **0.943** | 0.114 | -0.122 | -0.031 |
| EC | **0.922** | 0.097 | -0.201 | -0.051 |
| $K^+$ | 0.065 | -0.192 | **0.810** | -0.057 |
| Fe | -0.111 | 0.120 | **0.786** | 0.105 |
| $Ca^{2+}$ | 0.415 | **0.616** | 0.014 | -0.360 |
| $Mg^{2+}$ | **0.757** | 0.294 | 0.065 | -0.251 |
| $Na^+$ | **0.965** | 0.018 | -0.029 | -0.069 |
| $F^-$ | -0.298 | **0.630** | -0.248 | 0.439 |
| $Cl^-$ | **0.968** | -0.021 | -0.048 | -0.072 |
| $Br^-$ | **0.907** | 0.228 | 0.116 | -0.064 |
| $SO_4^{2-}$ | **0.884** | 0.090 | 0.172 | -0.103 |
| alkalinity | 0.254 | **0.771** | 0.016 | -0.055 |

The values bold demonstrate highly relationship in each component .
Rotation Method :Varimax with Kaiser Normalization.





**Table 4** Total variance explained of various factors .

| Component | Initial Eigenvalues | | | Extraction sums of squared Loadings | | | Rotation sums of squared loadings | | |
|---|---|---|---|---|---|---|---|---|---|
| | Total | %of Variance | Cumulative % | Total | %of Variance | Cumulative % | Total | %of Variance | Cumulative % |
| 1 | 6.486 | 49.892 | 49.892 | 6.486 | 49.892 | 49.892 | 6.132 | 47.166 | 47.166 |
| 2 | 1.602 | 12.325 | 62.217 | 1.602 | 12.325 | 62.217 | 1.596 | 12.277 | 59.443 |
| 3 | 1.262 | 9.708 | 71.926 | 1.262 | 9.708 | 71.926 | 1.448 | 11.142 | 70.585 |
| 4 | 1.096 | 8.434 | 80.359 | 1.096 | 8.434 | 80.359 | 1.271 | 9.775 | 80.359 |
| 5 | 0.715 | 5.503 | 85.863 | | | | | | |
| 6 | 0.629 | 4.840 | 90.702 | | | | | | |
| 7 | 0.485 | 3.733 | 94.436 | | | | | | |
| 8 | 0.340 | 2.618 | 97.053 | | | | | | |
| 9 | 0.180 | 1.387 | 98.441 | | | | | | |
| 10 | 0.097 | 0.749 | 99.190 | | | | | | |
| 11 | 0.054 | 0.415 | 99.605 | | | | | | |
| 12 | 0.033 | 0.257 | 99.862 | | | | | | |
| 13 | 0.018 | 0.138 | 100.000 | | | | | | |





**Table 5.** The levels of seawater intrusion with resistivity and EC values

| Level of seawater intrusion | Resistivity (Ωm) | EC (μs/cm) | Groundwater Facies |
|---|---|---|---|
| Extremely | < 5 | >1500 | Na-Cl |
| Moderately | 5-10 | 1000-1500 | Na-Cl, Ca-Na-Cl , Ca-Na-HCO3-Cl, Ca-Na-HCO3, Ca-HCO3-Cl |
| Slightly | >10 | <1000 | Ca-Na-Cl ,Ca-Na-HCO3, Ca-Na-HCO3-Cl, Ca-HCO3- Cl |




**Figure captions**
**Figure 1a.** Study area map including VES points, sample collected well and location of four cross-
section lines A-A', B-B', C-C', and D-D'.
**Figure 1b.** Land use map of the study area (Land Development Department, 2011).
**Figure 2.** Geological map in the study area (adapted from the Department Mineral Resources 2007).
**Figure 3.** Hydrological map in the study area (adapted from the Department of Groundwater Resources
(DGR), 2014).
**Figure 4.** Schlumberger configuration.
**Figure 5.** (**a**) Apparent resistivity type curve (type H) for three horizontal layers. (**b**) IPI2WIN
interpretation of VES St-25.
**Figure 6.** Pseudo cross-section lines A-A', B-B', C-C' and D-D'.
**Figure 7.** Resistivity value versus lithologic-log from the pseudo cross-section lines D-D' (well Q168
versus VES St 47 and well PW 7962versus VES St 34).
**Figure 8.** Geological cross-section lines as follows: (**a**) A-A'; (**b**) B-B'; (**c**) C-C'; and (**d**) D-D'.
**Figure 9.** Apparent resistivity map for AB/2 equals: (**a**) 5 meters; (**b**) 10 meters; (**c**) 30 meters; (**d**) 50
meters; (**e**) 70 meters; (**f**) 100 meters; (**g**) 150 meters; and (**h**) 200 meters.
**Figure 10.** Hydrochemical analysis of groundwater sample plotted in the piper diagram.
**Figure 11.** The relationship plotting between Na and Cl concentration (in meq/L) in the groundwater
samples collected from different aquifers.
**Figure 12.** Hydrochemical Facies Evolution Diagram (HFED) for depicting the salinization process in
this area.
**Figure 13.** A component plot in rotated space.
**Figure 14.** The boundary of seawater intrusion in the Qcl aquifer, based on the EC contour map
superimposed on the resistivity map.






























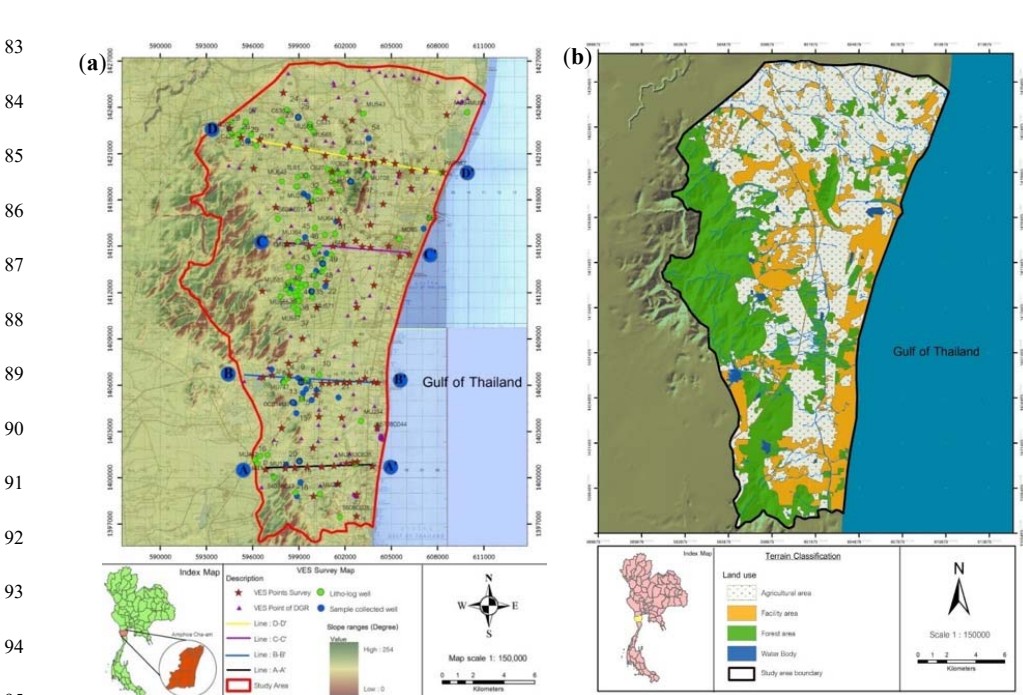

**Figure 1a.**

**Figure 1b.**



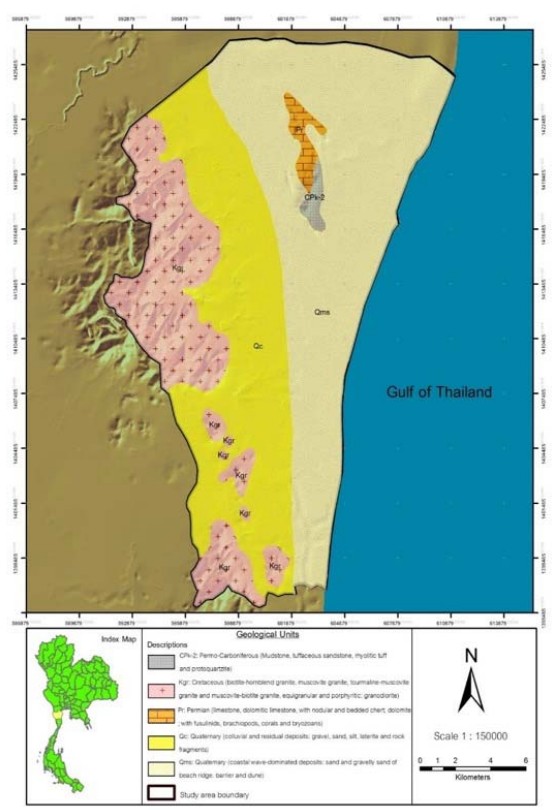


**Figure 2.**






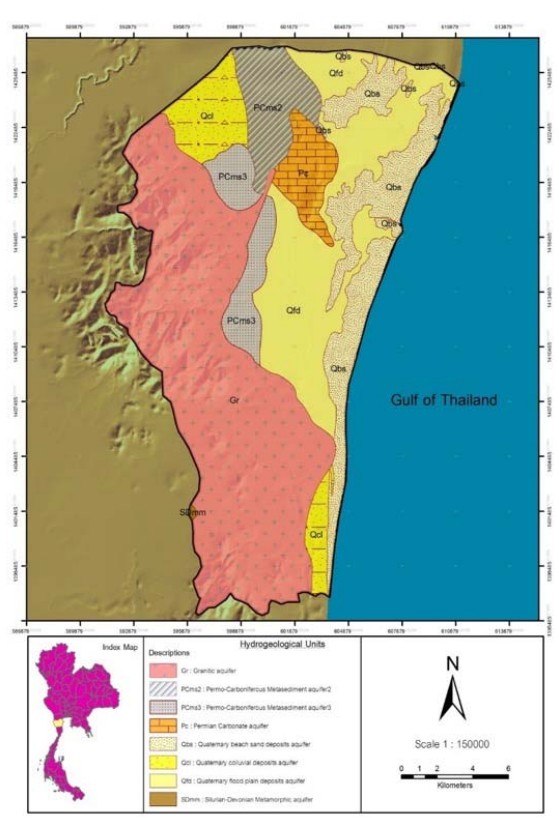


113                                **Figure 3.**










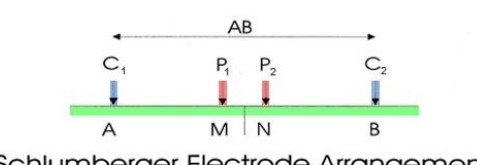

Schlumberger Electrode Arrangement


129                               **Figure 4.**




























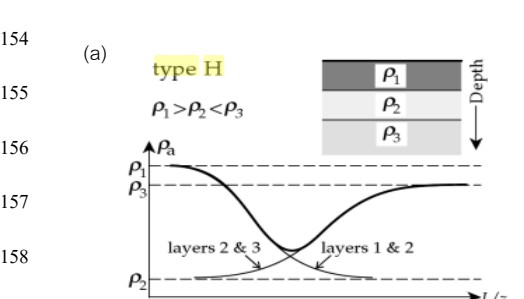 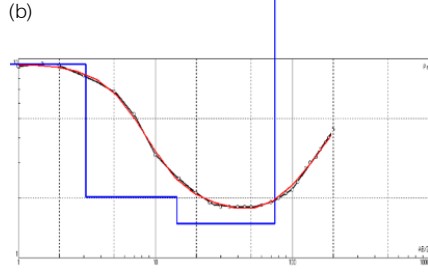






160                              **Figure 5.**





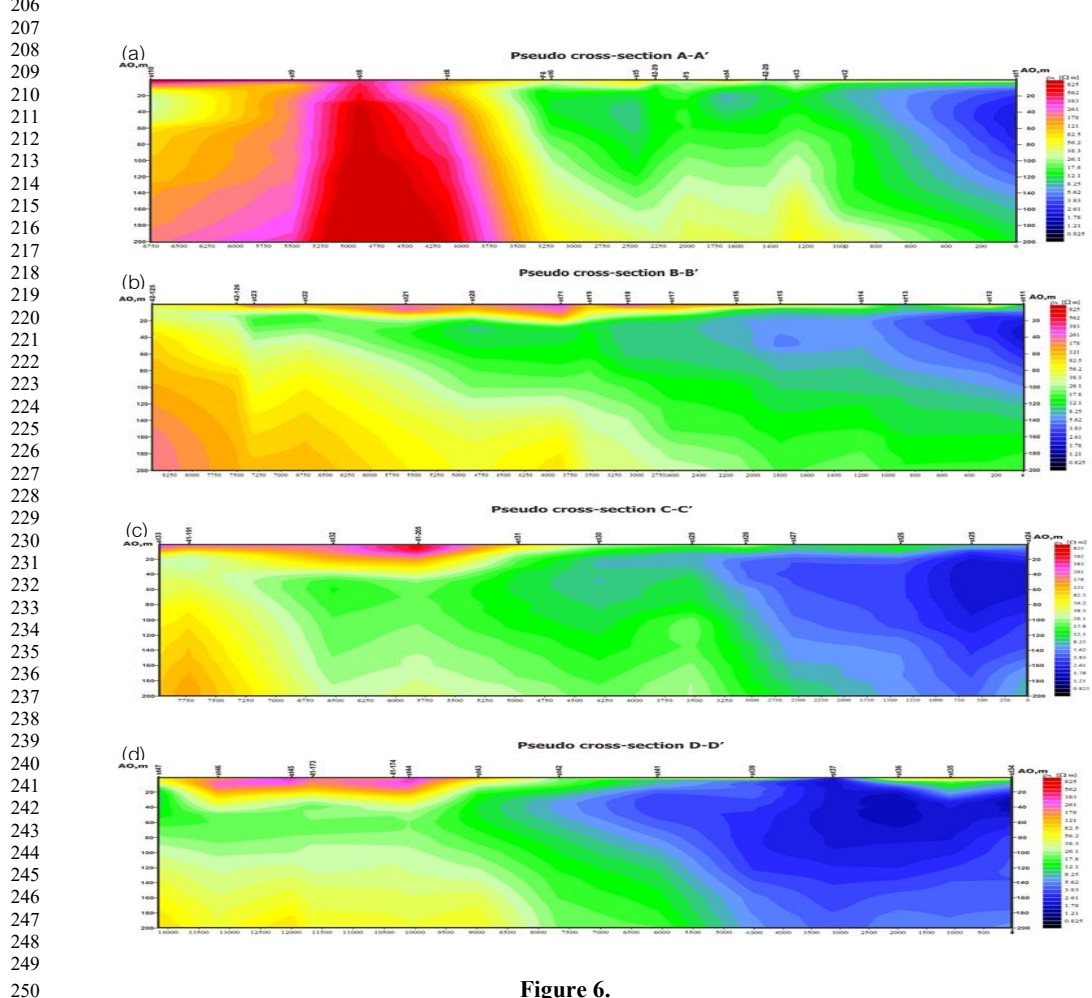

**Figure 6.**





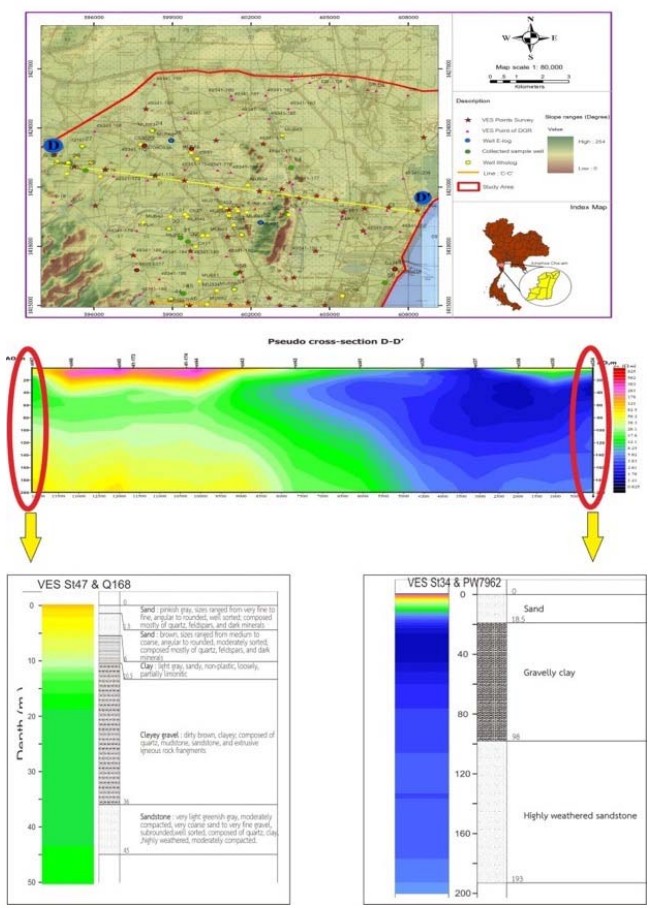


**Figure 7.**





























**Figure 8.**










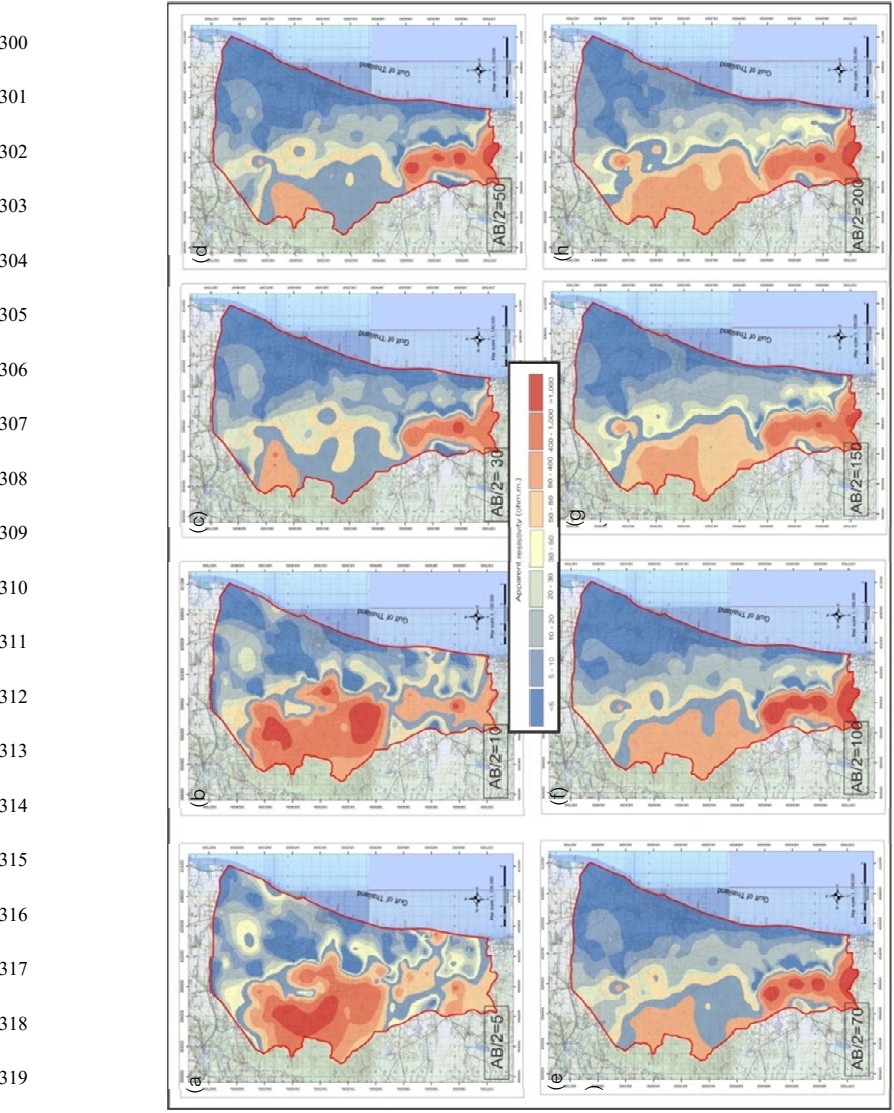

**Figure 9.**





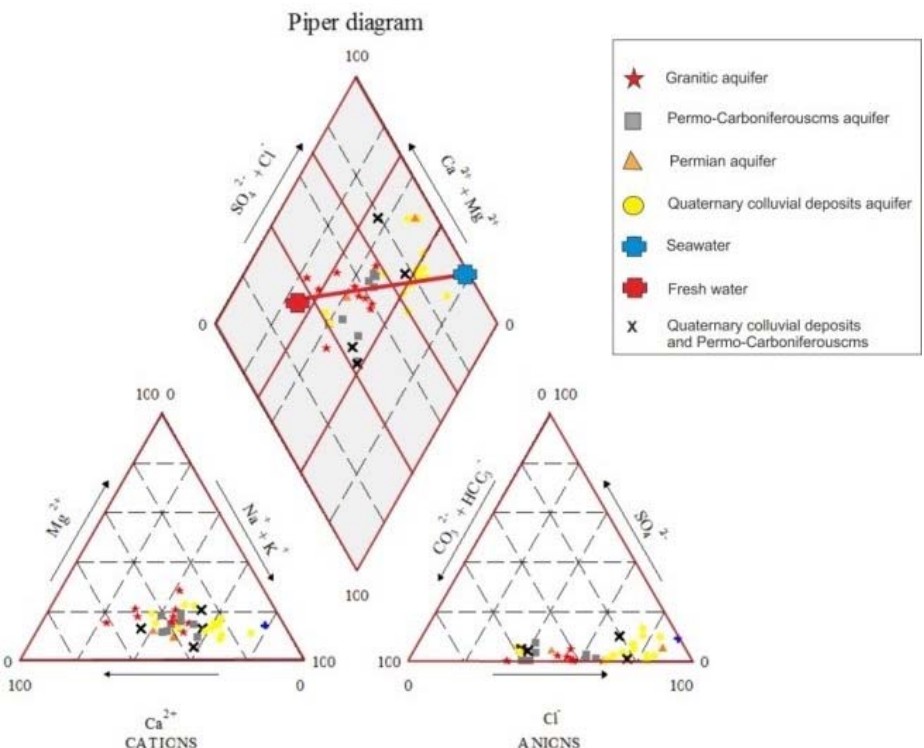


321                                        **Figure 10.**






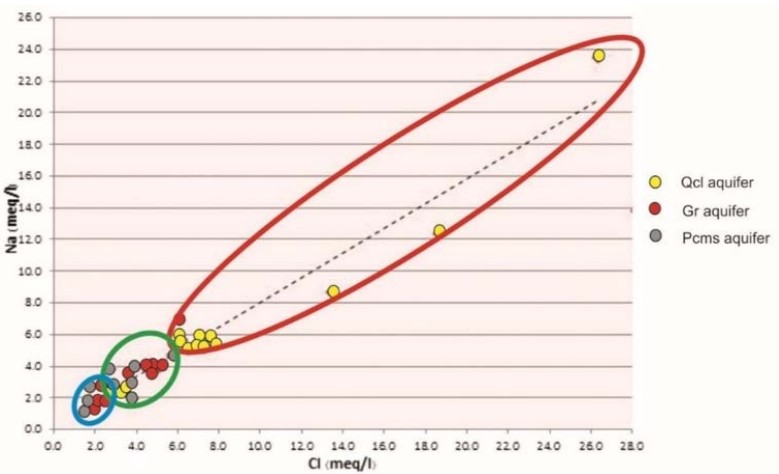


**Figure 11.**






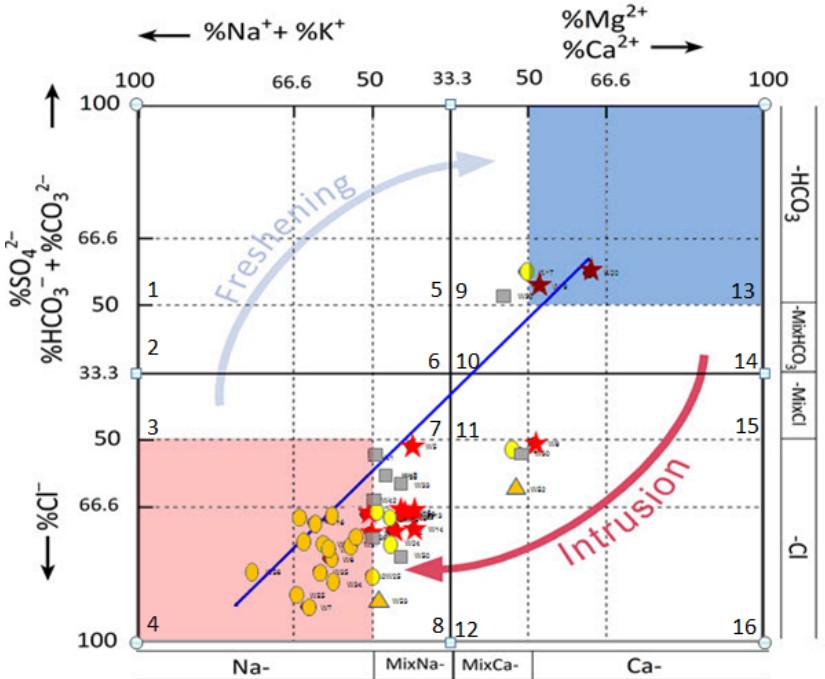



350                                          **Figure 12.**







**Figure 13.**

















**Figure 14.**
