# Peer review of "Assessment of seawater intrusion using multivariate statistical, hydrochemical and geophysical techniques in coastal aquifer, Cha-am district, Thailand"

_Hydrology and Earth System Sciences, 2018_

## Referee Comment (RC1) · A. Dell'Oca (Referee) · 5 Jul 2018

In the present work some interesting methodologies (based on geophysical and geo-chemical analysis) are applied in order to investigate the occurrence of Seawater intrusion. I personally find of interest the use and combination of diverse methodologies in order to analyze groundwater dynamics, but it is my understanding that in the current paper the level of novelty (both in terms of the development of the methodologies and/or combination of current ones in an innovative fashion) is poor. Moreover, the explanation of the concepts and principles at the base of the adopted methodologies

(here I refer to the geo-physical, geo-chemistry and statistical analysis) is insufficient, e.g., I struggled in understand the Principal Component Analysis and its benefits as presented even though I already had it in my background. Furthermore, the quality of the English is poor in several sentences, which makes hard to follow the work. The quality of the figure is very low, both in terms of graphics and completeness (e. g., what is along the axes in Fig. 5b? Where is the legend?) On top of these considerations, jointly with an unsatisfactory presentation of the ideas in the work I would recommend the work to be rejected for HESS and rearranged for another journal.

---

## Referee Comment (RC2) · Anonymous Referee #2 · 12 Jul 2018

Journal: HESS Title: Assessment of seawater intrusion using multivariate statistical, hydrochemical and geophysical techniques in coastal aquifers, Cha-am district, Thailand Author(s): Jiraporn Sae-Ju et al. MS No.: hess-2018-137 MS Type: Research article

I – General comments: In this manuscript, the authors discuss about a multidisciplinary approach to characterise seawater intrusion: coupling geophysical measurements and statistical analysis of the hydrochemical data. In fact, resistivity measurement through geophysics electrical methods are getting more and more popular to characterise seawater intrusion (SWI). As low resistivity can show the part of the aquifer affected by seawater. Therefore it can also be largely dependent on the bulk properties. Many studies can be found trying to compare electrical data with borehole sampling analysis. Therefore by adding multicomponent analysis it could give more details and can bring innovations. Therefore there are some important issues that require attention if addressed this would lead to the paper being suitable for publication. I recommend to rewrite the article with a simplification of the description of the geological and hydrogeological context. This part is hard to follow due to a large number of acronyms used. Then a better description of the methodology applied, even if the VES methodology is old it needs to be carefully described, in order to share the method for other applications, but also for the validation of the methodology. I would also recommend you to reduce the number locations map by grouping some of them and the quality of the figures must be improved. I will add that the innovative part of this paper, with the multicomponent analysis, is under interpreted and should be given more attention. The main import issue here is the English grammar that must be revised.

As a consequence, I would recommend that this paper should be rejected.

II – Specific comments

1. Abstract: acronyms are already used without defining what it is

2. Section 2.2, the hydrogeology part is a bit difficult to follow, there are a lot of formations (6-7 in total) and information linked to that one, it would be easier to put formation name, formation age, thickness, depth to water table, well yield and all over information in a table form.

3. Section 2.2, "the groundwater has accumulated in the pore..." this is very redundant and inherently assumed when we are talking about groundwater.

4. Section 3, Schlumberger acquisition has been largely described in the literature, you must cite older papers and grab the figure from : Dobrin, M. and Savit, C., Introduction

to Geophysical Prospecting, McGraw-Hill, New York, 867 p., 1988. Telford, W.M., Gel-dart, L.P. and Sheriff, R.E., Applied Geophysics, Cam-bridge Univ. Press, Cambridge, 770 p., 1990.

5. Section 3.1, the VES method must be explained in detail and more clearly, please have a look at some other articles using the same method, as in this example : Batte, A.G., Muwanga, A., Sigrist, P.W., Owor, M., 2008. Vertical electrical sounding as an exploration technique to improve on the certainty of groundwater yield in the fractured crystalline basement aquifers of eastern Uganda. Hydrogeology Journal 16, 1683–1693. https://doi.org/10.1007/s10040-008-0348-4

6. Section 3.2, nitric acidification of sample is done in order to avoid precipitation reactions inside the sampling bottle in order to keep it stable until the analysis.

7. Section 3.2, redundancy for anions and cations, just say once that chemical analysis was carried out for cations (which) and anion (which).

8. Section 3.2, piper diagram & HFE <– please cite in the literature (as it's done in section 4.4, Galloway and Kaiser, 1980)

9. Section 4.1, are you presenting the results for only one transects?

10. Section 4.1, what is it an H type curve? Acronyms are still used with abundance ("St25" ?)

11. Section 4.1, are you assuming a homogeneity of resistivity in the layers, so homogeneity of the geology inside the layer? What about changes of salinity inside the same layer? You should say ranges of resistivity.

12. Section 4.1, is there a correlation between the interpretation of all the points? You are not presenting it.

13. Section 4.1, is the measurement on those cross sections had been made at the same time-period?

14. Section 4.1, again the acronyms used a lot, VES points don't need to be cited while the location of those points are not presented on the maps.

15. Section 4.1, you can't make affirmations about the geology only by electrical measurement, you should validate with borehole cuttings.

16. Section 4.1, a VES point of measurement (as presented on figure 7) should be presented before the figure 6 to introduce the correlation between the other points in order to explain how the cross-correlation has been made.

17. Section 4.2, the title of the section is not appropriate because this is not "geoelectrical section", it's already the geological interpretation that you made from VES and from boreholes log data.

18. Section 4.2, the last sentence about the limited number of borehole log data should come at the beginning, in a way to justify the usage of VES but it could also be a limitation for the validation of the electrical data in the geological continuity of layers in the zones where you have few data.

19. Section 4.3, what is the location of the various areas cited in this section?

20. Section 4.3, what is AB/2 followed by the depth?

21. Section 4.4, software used must be cited in the methodology part than in the results.

22. Section 4.4, again usage of acronyms (line 332 : W5? ; line 335: W19 and W20 ?)

23. Section 4.4, be careful where you are using the word "influenced" for the influence of the seawater intrusion, do you think influenced or literally affected by it?

We agree with all these points that paper that must be revised.

III – Technical corrections

1/ This paper really needs a deep revision of the English grammar and syntax.